# Preventive Effects of Botulinum Neurotoxin Long-Term Therapy: Comparison of the ‘Experienced’ Benefits and ‘Suspected’ Worsening Across Disease Entities

**DOI:** 10.3390/jcm14020480

**Published:** 2025-01-14

**Authors:** Harald Hefter, Sara Samadzadeh

**Affiliations:** 1Department of Neurology, University of Düsseldorf, Moorenstrasse 5, 40225 Düsseldorf, Germany; sara.samadzadeh@yahoo.com; 2Charité–Universitätsmedizin Berlin, Corporate Member of Freie Universität Berlin and Humboldt-Unverstät zu Berlin, Experimental and Clinical Research Center, 13125 Berlin, Germany; 3Department of Regional Health Research and Molecular Medicine, University of Southern Denmark, 5230 Odense, Denmark; 4Department of Neurology, Slagelse Hospital, 4200 Slagelse, Denmark

**Keywords:** prevention, long-term outcome, botulinum neurotoxin therapy, natural history, disease progression, assessment of therapy

## Abstract

**Background:** Repetitive intramuscular injections of botulinum neurotoxin (BoNT) have become the treatment of choice for a variety of disease entities. But with the onset of BoNT therapy, the natural course of a disease is obscured. Nevertheless, the present study tries to analyze patients’ “suspected” course of disease severity under the assumption that no BoNT therapy had been performed and compares that with the “experienced” improvement during BoNT treatment. **Methods:** For this cross-sectional study, all 112 BoNT long-term treated patients in a botulinum toxin out-patient department were recruited who did not interrupt their BoNT/A therapy for more than two injection cycles during the last ten years. Patients had to assess the remaining severity of their disease as a percentage of the severity at onset of BoNT therapy and to draw three different graphs: (i) the CoDB-graph showing the course of severity of patient’s disease from onset of symptoms to onset of BoNT/A therapy, (ii) the CoDA-graph illustrating the course of severity from onset of BoNT/A therapy until recruitment, and (iii) the CoDS-graph visualizing the suspected development of disease severity from onset of BoNT/A therapy until recruitment under the assumption that no BoNT/A therapy had been performed. Three different types of graphs were distinguished: the R-type indicated a rapid manifestation or improvement, the C-type a continuous worsening or improvement, and the D-type a delayed manifestation or response to BoNT therapy. Four patient subgroups (cervical dystonia, other cranial dystonia, hemifacial spasm, and the migraine subgroup) comprised 91 patients who produced a complete set of graphs which were further analyzed. The “experienced” improvement and “suspected” worsening of disease severity since the onset of BoNT/A therapy were compared and correlated with demographical and treatment related data. **Results:** Improvement was significant (*p* < 0.05) and varied between 45 and 70% in all four patient subgroups, the “suspected” worsening was also significantly (*p* < 0.05) larger than 0, except in the migraine patients and varied between 10 and 70%. The “total benefit” (sum of improvement and prevented “suspected” worsening) was the highest in the other cranial dystonia group and the lowest in the migraine subgroup. The distributions of R-,C-,D-type graphs across CoDB-, CoDS-, and CoDB-graphs and across the four patient subgroups were significantly different. **Conclusions:** (i) Most BoNT long-term treated patients have the opinion that their disease would have further progressed and worsened if no BoNT/A therapy had been performed, (ii) The type of response to BoNT/A is different across different subgroups of BoNT/A long-term treated patients.

## 1. Introduction

Botulinum neurotoxins (BoNTs) are among the most toxic molecules known thus far [1,2,3]. Nevertheless, BoNT injections have become a safe and effective therapy and an excellent remedy for a variety of diseases (such as migraine [4,5,6], spasticity in enfants [7] and adults [8], hypersalivation [9], axillary [10] and forehead [11] hyperhidrosis, hemifacial spasms [12]) and are even a treatment of choice for patients with focal dystonia [6]. BoNT injection therapy can be performed over decades without relevant side effects [6], and, for most disease entities, a stable level of improvement can be maintained if the BoNT injections are applied repeatedly [13] and an adequate adaption of a dose and injection scheme to a possible disease progression is performed [13]. Although BoNT therapy is not a causal but a pure symptomatic treatment, the majority of patients are satisfied with this treatment [14] and it has a high adherence to therapy [15].

Long-term outcomes depend on a variety of factors, such as disease entity and dose per session [16], but also on initial disease severity [17] and natural course of the disease before onset of BoNT therapy [18]. It could be demonstrated that in patients with cervical dystonia (CD) and a low severity of CD at onset of BoNT therapy, CD may become worse despite the BoNT injection therapy with standard doses (for details see [17]). Furthermore, it has been reported that the long-term outcome of BoNT therapy in CD is inversely correlated to time to BoNT therapy, as has been reported in long-term outcomes in primary dystonia after deep brain stimulation (DBS) operations [19,20]. This underlines the importance of an early onset of BoNT therapy in the course of a disease and the application of a sufficiently high dose even when the severity of the disease is not yet fully developed.

Time to therapy heavily depends on the interference of patient’s disease with their everyday life activities and occupational demands, which also influences adherence to therapy and long-term outcome. This becomes obvious when very different disease entities are compared, such as focal hand dystonia in musicians [21] and spasticity of extremities in post-stroke patients [15,16]. When a special disease becomes clinically manifested and the patient looks for help, at least two aspects become highly important for the patient: (i) to what extent his disease will further develop and (ii) what benefit can be expected when a special therapy is initiated. The benefit of a therapy will therefore be a combination of an experienced improvement and the prevention of a possible worsening. However, when a continuous intervention therapy, such as the BoNT injection therapy, has been initiated, the natural course of a disease cannot be observed anymore, but the patient’s fears and sorrows regarding whether or not they suffer from a further developing disease may persist.

This was analyzed in a recent pilot study [18] on 27 CD-patients by comparing the improvement or benefit (EBEN) of BoNT therapy with the “suspected” worsening (SWORS) of CD under the assumption that no BoNT injections or DBS-operation had been performed. On the one hand, a worsening of 40 to 60% was suspected in comparison to the severity of CD at onset of BoNT therapy. This “suspected” worsening appeared to be realistic since CD-patients with a long time to therapy and a high initial severity usually present with TSUI-scores [22] between 14 and 16 [17], which is much higher than the mean initial TSUI-score between 8 and 10 in the majority of de novo CD-patients (for details see [16,17]). On the other hand, these CD-patients also experienced a considerable improvement of the severity of CD of about 50%. Thus, BoNT therapy not only prevented a considerable worsening of the disease but also caused a significant improvement. In this pilot study, patients assessed the efficacy of BoNT therapy by rating the remaining severity of their disease in % of the initial severity at onset of BoNT therapy and by drawing the course of disease severity (CoD-graphs; for details see [18]).

The question remained whether the results of this pilot study indicating a preventive aspect of long-term BoNT therapy could be confirmed in a larger cohort of CD-patients and in other disease entities. To analyze this preventive aspect and the interplay between improvement and “suspected” worsening in more detail, we again used the method of course of disease (CoD) drawing in combination with patient’s assessment of the benefit of BoNT therapy.

## 2. Materials and Methods

This is a non-invasive, observational cross-sectional study. Inclusion into or exclusion from this study did not have any influence on the treatment of the patients. According to our local ethics committee, there is no need for a special application or approval of such a type of study.

### 2.1. Patients

Inclusion criteria for the present study were: (i) diagnosis of disease confirmed in the outpatient department of the department of Neurology of the University Düsseldorf, (ii) continuous BoNT treatment for at least one year. Excluded were: (i) patients with cessation of BoNT treatment for more than 6 months (corresponding to two treatment cycles), (ii) patients with memory problems as, e.g., Alzheimer’s disease, (iii) patients with treated psychiatric disorders, e.g., moderate to major depression, (iv) patients with other severe disabling disorder, e.g., inflammatory diseases or severe orthopedic problems, and (v) patients with a combination of BoNT treated diseases (e.g., CD plus hemifacial spasms (HFS)).

Patients were informed on the purpose of the present study while they were waiting for their next routine botulinum neurotoxin type A (BoNT/A) injection. After patients had given informed consent, they received their BoNT injection and were recruited for the present study.

Between 5 June 2023 and 10 September 2023, 141 patients were screened. Eighteen patients were excluded because of too low adherence, four patients because of psychiatric disorders, three patients because of other disabling diseases, and four patients because of a combination of BoNT treated diseases. Finally, 112 patients were recruited, assessed for the treatment effect, and three CoD-graphs were created (see below).

### 2.2. Assessment of Disease Severity and Course of Disease Drawing

After the routine BoNT/A injection, patients were asked at what age they had noticed the first symptoms of their disease (AOS). Then they had to assess the actual remaining disease severity (RS-%) and the maximal “suspected” severity under the assumption that no intervention (neither BoNT-therapy nor DBS-operation) had been performed (SS-%). Both RS-% and SS-% had to be assessed as a percentage of the severity of symptoms at onset of BoNT-therapy, which was set to 100%.

Then, they had to draw (i) CoD severity from onset of symptoms to onset of BoNT-therapy (CoDB-graph), (ii) CoD severity from onset of BoNT-therapy to recruitment (CoDA-graph), and (iii) the suspected CoD severity from onset of BoNT therapy until recruitment (CoDS-graph) under the assumption that no intervention (neither BoNT-therapy nor DBS-operation) had been performed. Details of CoD-graph drawing have been described previously ([18]; see also Figure 1).

### 2.3. Classification of the Graph Type

For further analysis, a complete set of CoDB-, CoDS-, and CoDA-graphs were available for 91 patients (39 patients with idiopathic cervical dystonia (CD-group); 18 patients with other cranial dystonia (blepharospasm, Meige syndrome, oromandibular and oropharyngeal dystonia; CRD-group); 25 patients with hemifacial spasms (HFS-group); 9 patients with chronic migraine (MIG-group)). For each of the 273 CoD-graphs, the type of graph was determined.

A CoDB-graph was classified as R-type (rapid manifestation type) when 75% of the graph was lying above the straight line from the left lower corner (corresponding to the onset of symptoms) to the right upper corner (=100%) of the presented template into which the CoDB-graph had to be drawn (see Figure 1A). If 75% of the graph was lying under this line, the graph was classified as D-type (delayed manifestation type). All the rest of the graphs were classified as C-type graphs (continuous type) since, in this case, CoD indicated a continuous worsening.

For classification of the CoDA-graphs, a straight line was drawn from the left upper corner to the point where the CoDA-graph touched the right vertical of the given template (RS-D). When 75% of the CoDA-graph was lying below this line, the graph was classified as R-type (rapid response type), when 75% was lying above this line, the graph was classified as D-type (delayed response type). All other graphs were classified as C-type (continuous response type) (see Figure 1C).

If a CoDA-graph indicated a clear systematic worsening after an initial good response, the patient was classified as secondary non-responder (STF). Patients in whom a secondary worsening could be compensated by a dose increase were classified as patients with a “pseudo”-STF (P-STF). If the graph indicated an improvement of less than 20% at any time of the treatment, the patient was classified as primary non-responder (PTF).

For classification of the CoDS-graphs, a straight line was drawn from the left upper corner to the point where the CoDS-graph touched the right vertical (SS-D) of the given template (see Figure 1B). When 75% of the CoDS-graph was lying above this line, the graph was classified as R-type (rapid worsening type), when 75% was lying below this line, the graph was classified as D-type (delayed worsening type). All other graphs were classified as C-type graphs (continuous worsening type).

### 2.4. Demographical, Treatment-Related, and Long-Term Outcome Data

For further analysis, the following parameters were extracted from the charts of the patients: age at recruitment (AGE), age at onset of BoNT therapy (AOT), actual severity of CD assessed by the treating physician by means of the TSUI-scale (ATSUI; which scores whether head flexion/extension, tilt to the right or left or head rotation are permanently present or not, and whether a head tremor is present or not [22]), actual used BoNT/A preparation, and actual total dose of BoNT/A per session (ADOSE). For sake of comparison, doses were transformed into unified dose units (uDU) by dividing abobotulinum neurotoxin type A (aboBoNT/A) doses by three and leaving onabotulinum neurotoxin type A (onaBoNT/A) and incobotulinum neurotoxin type A (incoBoNT/A) doses unchanged (following a consensus paper [23]). Calculated were duration from onset of symptoms to onset of BoNT therapy (DURS = AOT-AOS) and duration of therapy (DURT = AGE-AOT).

### 2.5. Statistics

Final statistical analysis was performed on 91 patients (see Section 3). The data of 21 patients were excluded from the final analysis: the number of patients with focal dystonia of the extremities (*n* = 5; 4 patients with writer’s cramp, 1 patient with foot dystonia), of patients with neurodegenerative disorders (*n* = 4; one patient with corticobasal degeneration, 3 patients with idiopathic Parkinsonian symptoms who were treated because of hypersalivation), and of the patients with generalized dystonia (*n* = 2) as the data were too small for a statistical analysis. The group of patients with spasticity (*n* = 8; SPAS-group) was highly heterogeneous (5 patients with stroke, 1 patient with severe affection of the cervical spine, 2 patients with cerebral palsy) and therefore were also excluded from further analysis. Two further patients did not succeed in producing continuous graphs: the eldest patient of the entire cohort had impaired vision, and the other patient suffered from a hand tremor.

Differences in the distributions of the three graph types (R-, C-, D-type) across CoDB-, CoDS-, and CoDA-graph drawing as well as across the four patient subgroups (CD, CRD, HFS, MIG) were analyzed by means of chi^2^-testing. Whether the distributions of graph types of CoDB-, or CoDS-, or CoDA-graphs were different within a patient group was also analyzed by chi^2^-testing. Descriptive statistics (mean values, standard deviations) were determined for the following parameters: AOS, AOT, AGE, DURS, DURT, RS-%, RS-D, SS-%, SS-D, ADOSE. For patients with CD, the mean ATSUI was also calculated. Improvement (EBEN-% = 100 − (RS-%), EBEN-D = 100 − (RS-D)) and “suspected” worsening (SWORS-% = (SS-%) − 100, SWORS-D = (SS-D) − 100) determined from the assessment of the patients or from drawing the CoDA- or CoDS-graph were added to yield the total benefit (TBEN-%, TBEN-D).

The H-test was used to analyze whether improvement (EBEN-%, EBEN-D) and “suspected” worsening (SWORS-%, SWORS-D) were significantly larger than 0.

All statistical procedures were part of the R software statistics package (version 4.3.1). The dplyr package was utilized for data manipulation, the ggplot2 package for data visualization (histograms, box plots).

## 3. Results

### 3.1. Differences in Demographical, Treatment-Related, and Long-Term Outcome Data

In Table 1, the mean values (MV) and standard deviations (SD) of the demographic, treatment-related, and long-term outcome data, as well as the suspected worsening without BoNT therapy are presented for four patient subgroups.

Patients with migraines (MIG-group in Table 1) were all females, had the youngest age (53.0 years in the mean), had the longest time to therapy (24 years in the mean!), were exclusively treated with onabotulinumtoxin (onaBoNT/A), and reported the best highly significant (*p* < 0.0001) relative mean improvement (100 − (RS-%)) of more than 65% and suspected only a mild non-significant (*p* > 0.05) further worsening ((SS-%) − 100 = SWORS-%) of less than 10% in the mean beyond the level of severity at onset of BoNT therapy.

Patients with hemifacial spasms (HFS-group in Table 1) were the eldest patients (mean age: 73.7 years), had the shortest time to therapy (mean DURS: 2.9 years), were treated with all three BoNT/A preparations with the lowest dose of about 47 uDU in the mean. These patients also reported a significant (*p* < 0.001) relative improvement (EBEN-%) of about 60%, but also suspected a significant (*p* < 0.05) further worsening of about 40%.

Patients with cranial dystonias other than CD (CRD-group in Table 1) were the eldest patients at onset of therapy, had an equally short duration to therapy as the HFS-patient, reported a significant (*p* < 0.01) relative improvement of 55% in the mean, but suspected the highest possible, significant (*p* < 0.001) worsening without BoNT therapy of more than 60%. Two patients experienced a secondary worsening after an initial good response (STF) which could at least partially be compensated by an increase in dose. Therefore, these two patients were classified as patients with “peudo” secondary treatment failure (P-STF).

Patients with CD (mean age: 67.5 years) had the longest duration of treatment (mean DURT: 14.7 yrs) and were treated with the highest doses (mean dose: 337 uDU). In two patients, a partial secondary treatment failure (PTF) was observed, and in two further patients, a P-STF, and in eight patients (=20.5%), an antibody-induced STF was observed. The majority of patients were treated with incoBoNT/A because of its lower antigenicity compared to abo- or onaBoNT/A [24]. Patients with CD reported a significant (*p* < 0.05) improvement of more than 47%, as well as a highly significant (*p* < 0.001) suspected worsening of more than 50%. Mean TSUI-score [22], which estimates long-term outcome of BoNT therapy in CD, was low (3.7) compared to previously reported TSUI-scores (4.8; [25]).

### 3.2. Differences in Improvement and “Suspected” Worsening in Four Patient Groups

In Figure 2, for each of the four patient groups (CD-, CRD-, HFS-, and MIG-group) the “experienced” improvement or benefit drawn by the patients (EBEN-D = 100 − (RS-D); light gray columns in Figure 2) are compared to the suspected worsening (SWORS-D = (SS-D) − 100; dark gray columns in Figure 2) and to the total benefit (TBEN-D = (EBEN-D) + (SWORS-D) = (SS-D) − (RS-D); black columns in Figure 2). The highest TBEN-D is observed in the CRD-group and the lowest in the MIG-group.

### 3.3. Differences in the Shape of the CoD-Graph Drawings

As explained in the methods for each of the CoDB-, CoDS-, and CoDA-graph categories, three types of severity drawing were distinguished: (i) the type indicating a rapid worsening or improvement (R-type), (ii) the type describing a continuous development of the disease either worsening or improvement (C-type), and (iii) the type indicating a delayed worsening or improvement (D-type) (Table 2).

In Figure 3, the distributions of the R-, C-, D-types across the CoDB-, CoDS-, and CODA-graph categories are presented. Chi^2^-testing (chi^2^(4) = 60.548; *p* < 2.225 × 10^−12^) clearly demonstrates that the patients did not choose the shape of the CoD-graphs deliberately, but significantly modified the shape of the graph depending on the task and whether they had to draw CoD before or after BoNT therapy or whether they had to anticipate how CoD would have developed without BoNT therapy. The most frequent CoDB-graph type was the D-type, the most frequent CoDS-graph type was the C-type, and the most frequent CoDA-graph type was the R-type (Figure 3).

In Figure 4 the distributions of the R-, C-, D-types are presented for the four patient groups. Chi^2^-testing (chi^2^(6) = 29.848; *p* < 4.201 × 10^−5^) showed that the distributions of the graph types produced by the four different patient subgroups were significantly different (Figure 4). In all four patient groups, the C-type was the most frequently drawn type of CoD-graphs. But in the HFS-group much more R-graphs were drawn than D-graphs, whereas in the CRD- and MIG-group more D-graphs were drawn than R-graphs.

In Figure 5 the distributions of the R-, C-, D-types are presented for the CoDB-, CoDS-, and CoDA-graphs within the four patient groups. When the distributions of the R-, C-, and D-type were compared across patient groups, neither for the CoDB-graphs (chi^2^(6) = 12.51; *p* = 0.05151) nor for the CoDS-graphs (chi^2^(6) = 3.2274; *p* = 0.7798) were significant differences were found. However, for the CoDA-graphs, highly significant differences were detected (chi^2^(6) = 74.742; *p* < 4.337 × 10^−14^).

## 4. Discussion

### 4.1. The Long-Term Improvement Assessed by Patients in 4 Different Patient Subgroups

In the present study, the primary outcome measure is patient’s assessment of the remaining severity of his/her disease in % of the disease severity at the onset of BoNT therapy (=100%) either given verbally when asked (RS-%) or documented on a 10 cm visual analog scale (VAS) when the CoDA-graph had to be produced (RS-D). Therefore, the outcome in the present study can be compared with the outcome in other studies [15,16] in which a VAS- or other global impression scale is used.

A recent Cochrane meta-analysis for BoNT therapy of chronic migraines [26] reports an improvement of 30% (3 cm on a 10 cm VAS) of disease severity, which is much lower than the 67% observed in the present study. But that analysis did not take into account the influence of repetitive injections, the duration of therapy, and that treatment efficacy may increase with duration of treatment as reported by more than 40% of our MIG-patients. Therefore, long-term BoNT therapy of chronic migraines probably yields much better outcomes than suggested by this Cochrane review.

In patients with HFS BoNT, long-term therapy is highly effective. In a recent survey by Park et al. [12], it was reported that improvement rates were between 78% and 98% for five studies with more 100 patients using either a VAS, a 0–4-point rating scale, or subjective assessment, as in the present study, and two other scales. HFS Lee et al. [15] also reported an improvement of about a 77% reduction in symptoms. At first glance, this outcome appears to be much better than in our study. But our patients did not assess the peak effect of an injection at week 4 but rather the outcome at the end of an injection cycle when the effect of the previous injection had already declined and they received their next BoNT/A injection.

The outcome in patients with blepharospasm, Meige-syndrome, or other cranial dystonias, such as oromandibular or oropharyngeal dystonia (CRD-group), varies from patient to patient considerably. This has previously been reported by several centers [27,28,29,30], and seems to be very much dependent on the pattern of involuntary muscle activity [31]. In a mixed population (like ours), an overall improvement of 50% [27] and 63% [32] was observed, which nicely matches the 55% improvement in the present study. Lee et al. [15] reported an even better reduction in symptoms of 64%, but also reported a high drop-out rate in contrast to the long duration of BoNT therapy in the present study (see Table 1).

Also in CD, the response to BoNT therapy varies considerably. Our patients who rate the remaining severity of CD at the end of an injection cycle experienced an improvement of about 50%. In Lee et al. [15], a reduction in symptoms of about 68% is reported. Also, in CD, the peak effect at week 4 is better than the effect after 12 weeks at the end of a treatment cycle. This explains why, in many trials, the efficacy of BoNT therapy appears to be better than 50% (for a review see [33]). The present patient assessment of long-term outcome after BoNT therapy of CD of 49% is in full agreement with physician’s rating of the severity of CD by means of the TSUI-score [22]. In 221 BoNT long-term treated CD-patients, the TSUI-score improved from a baseline value of 9.71 to 4.81 (50.4%) after a treatment duration of more than five years [25].

Regardless of whether patients had to assess the remaining severity verbally (RS-%) or by drawing on a VAS-scale (RS-D), the improvement varied between 45% and 70% across all four patient groups. Time from onset of symptoms to BoNT therapy considerably varied between 2.9 years in the HFS-group to 24.3 years in the MIG-group. This time to therapy (DURS) depends, on the one hand, on the handicap and/or stigmatization resulting from the underlying disease and on the license situation on the other. Involuntary movements of facial muscles are very stigmatizing. This is most likely the reason why patients with HFS or blepharospasm or Meige syndrome or other cranial dystonias (CRD-group) had the shortest DURS around 3 years. OnaBoNT/A was licensed for the treatment of migraine in Germany in 2011 [4,5] and only for patients with chronic migraines, which usually is the end-stage of long pain carrier [24]. This is the reason why patients with migraines had such a long time to therapy.

### 4.2. Differences in the Distributions of R-, C-, and D-Type Across Different Disease Entities

The drawing of the CoDA-graphs visualizes patients’ assessments of the response to BoNT therapy. The highly significant difference in the distributions of R-, C-, and D-types of CoDA-graphs demonstrates clearly that the different patient groups respond differently to the BoNT therapy and that the shape of the CoD-graphs contains relevant information. Overall, the majority of patients reported a continuous or even delayed response (Figure 3). But many patients experienced a rapid improvement corresponding to a R-type drawing. The majority of patients with HFS (18/25 = 72%) and many of the CD-patients (18/39 = 46%) had drawn a R-type CoDA-graph thus indicating a rapid improvement after onset of BoNT therapy. These two patient groups had the longest duration of therapy (DURT; see Table 1), between 13 and 15 years in the mean without longer interruptions indicating a very high adherence to therapy. This has already been observed in a previous study comparing different disease entities [15].

Deliberate drawing of the course of disease and random assessment of the treatment effect by the patients would not have led to significant differences between different disease entities. However, we think that patients took the task to draw CoD-graphs seriously and assessed the treatment effect precisely. Therefore, not only are differences in the long-term outcome across different disease entities, but also different shapes of the CoD-graphs were produced. In our opinion, the patients’ drawings of the CoD provide relevant information for the treating physician.

We have used the 75%-criterion for the classification of the CoD-graphs. Since it has been the first time that the different CoD-graphs are analyzed in detail, the usefulness of this criterion has to be substantiated by further studies on CoD-graph drawing.

### 4.3. The “Suspected” Worsening Assessed by Patients in Four Different Patient Subgroups

The second parameter which was of main interest in the present study is the assessment of the “suspected” worsening of the underlying disease. Patients had to speculate about the development of their disease during the time from onset BoNT therapy until recruitment into the present study and had to assess the “suspected” severity of their disease (SS-%, SS-D) at the day of recruitment under the assumption that no BoNT therapy or DBS-operation had been performed.

No patient suspected a clear improvement of more than 10% without therapy. Most patients suspected a clear further worsening with the exception of the patients with migraine. The reason is that patients with chronic migraine had the impression that they had been on a final plateau of their disease development at onset of BoNT therapy. Some patients even indicated that they would have died by suicide if their pain syndrome would have progressed further. The variability of SS-% or SS-D across patient groups was higher (SS-%: 109 to 165%; SS-D: 111 to 170%) than that of RS-% or RS-D (see Table 1).

The ”suspected” worsening of about 40 to 60% in the CD group appears to be realistic. In our institution, the majority of patients present with an initial TSUI-score between 8 and 10 [16,17,18,32]. Only 25% of the de novo-patients present with a TSUI-score > 10 [17]. De novo-CD-patients with a TSUI-score > 16 are exceptional cases. Therefore, we think that the “suspected” worsening between 40 and 60% of the patients in the present CD-group meets the severity of CD in very severely affected de novo-CD-patients [17,18].

That patients with CRD suspect that their disease may progressively worsen also seems to be realistic. In some patients with Meige syndrome or OMD or oropharyngeal dystonia disease manifestation starts with hyperactivity in the external ocular muscles. In other patients only the ocular problems or movements of the mandibula are BoNT treated although tongue or swallowing problems are present. These patients live with the fear that their oropharyngeal or laryngeal problems may progress and may lead to a severe handicap.

Most of the patients with HFS know which of their facial muscles are involved in their disease process and are informed which muscles might cause further problems. This we think is the reason why “suspected” worsening was lower in the HFS-group compared to that in the CD- or CRD-group.

### 4.4. The Role of Disease Complexity for “Experienced” Improvement and “Suspected” Worsening?

There is no precise definition of disease complexity. However, in botulinum toxin therapy one aspect of complexity is whether a patient with a special disease entity is difficult to inject or not or can be injected without disturbances of relevant residual motor function (as in hand dystonia, e.g., [21]). In migraine the license studies recommend the injection at 31 points which can be targeted easily [4,5]. As long as the neck muscles are injected with low doses BoNT mediated side effects of BoNT therapy in migraine are mild, and the withdrawal rate is low (3%) [26].

Also, most facial muscles being affected in hemifacial spasm can easily be reached without special guidance technique [27]. Ptosis or hanging corner of the mouth can be avoided by the use of low doses for the eye lid and the zygomaticus muscle [27,31,32].

The treatment of CD is easy as long only superficial muscles are injected as the sternocleidomastoid muscle (SCM), the splenius capitis muscle (SPL), levator scapulae muscle (LSC), or the medial and posterior scaleni muscles (SCAL). But the injection of anterior and posterior deep neck muscles is more difficult and may afford special guidance techniques. Neck weakness and slowing problems may occur so that deep brain stimulation is an alternative to BoNT therapy in these patients.

In patients with Meige syndrome or other cranial dystonia the injection of jaw opening or tongue muscles may be necessary which are difficult to inject [29,30,31].

Thus, the “suspected” worsening (SWORS-D) is correlated with this aspect of complexity (Mig: 11.2%; HFS: 37.6%; CD: 62.4%; CRD: 70%) as is the case for the total benefit (TBEN-D) (Mig: 80.0%; HFS: 96.3; CD: 109.4%; CRD: 122.0%) although the benefit is inversely correlated with disease complexity (Mig: 68.8%; HFS: 58.8%; CD: 47.0%; CRD: 51.9%).

We therefore think that disease complexity and whether a BoNT preparation is licensed for early or only for late phases of a disease entity are more relevant for explaining the “suspected” worsening than loss of memory for details of disease severity with duration of BoNT therapy as hypothesized earlier [18]. Most patients remember fairly well the symptoms and handicaps they had when BoNT therapy was initiated.

## 5. Conclusions

Patients undergoing long-term BoNT treatment do not only experience a benefit between 50 and 70% in the mean. They also have the impression that BoNT has prevented a further worsening of their disease between 10 and 70%. This preventive aspect of BoNT therapy is probably a major reason for the excellent compliance observed in BoNT therapy.

## 6. Strengths and Limitations of the Study

The strength of the present study is that it is the first study which compares the “suspected” change in disease severity in BoNT long-term treated patients under the assumption that no BoNT therapy had been initiated with the benefit under BoNT long-term therapy across different disease entities. The present analysis of drawing the course of disease severity in different disease entities clearly showed that patients do not randomly produce different graph types and that this method yields relevant information for the treating physician. Furthermore, this study yields hints for a preventive aspect of BoNT therapy.

The present approach cannot be extended to all patients treated with BoNT. Patients with stroke, traumatic brain or spinal cord injury produce CoDB-graphs, which do not follow the R-, C- or D-type.

Unfortunately, we did not have access to patients who could have been used for a control arm to analyze the natural history of different disease entities in more detail. And there are no other studies available this study can be compared with. We therefore recommend to perform further studies on the possible progression of a disease without BoNT therapy and whether the results on differences in “suspected” worsening in different disease entities are realistic or not. To improve comparability with other studies, such a study should follow a prospective design with inclusion of data from standard scales as quality of life or VAS or global impression scales which are lacking in the present study

## Figures and Tables

**Figure 1 jcm-14-00480-f001:**
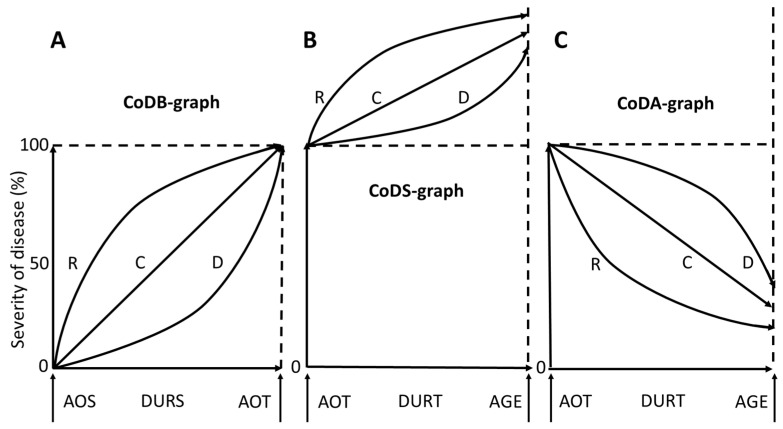
(**A**): Three different types (R-, C-, D-type) of graphs are presented indicating the course of disease severity before BoNT-therapy (CoDB-graphs) during the time span (DURS) between onset of symptoms (AOS) to onset of BoNT therapy (AOT). Severity of the disease at onset of BoNT therapy was used as reference (=100%). (**B**): Three different types (R-, C-, D-type) of graphs indicate the “suspected” course of disease severity under the assumption that no BoNT therapy had been performed (CoDS-graphs) during the time span (DURT) between onset of BoNT therapy (AOT) to the day of recruitment (AGE). (**C**): Three different types (R-, C-, D-type) of graphs indicate the “experienced” course of disease severity (CoDA-graphs) during the time span (DURT) between onset of BoNT therapy (AOT) to the day of recruitment (AGE).

**Figure 2 jcm-14-00480-f002:**
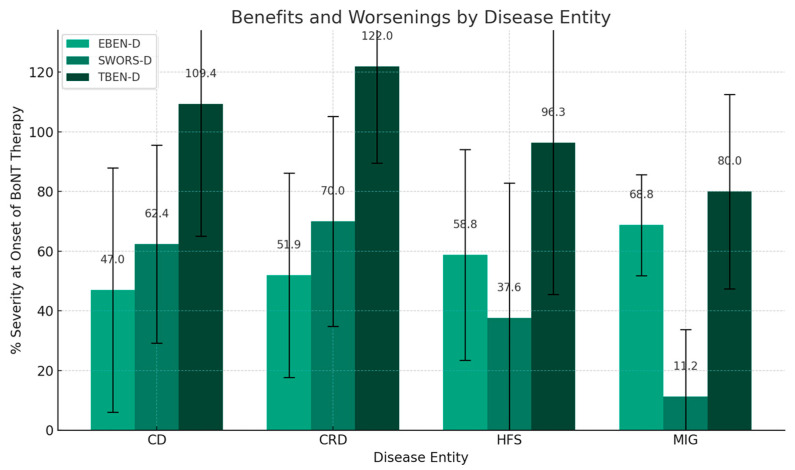
Comparative analysis of treatment impact across disease entities. This bar chart represents the mean values of benefit (EBEN-D; left light gray bar), “suspected” worsening (SWORS-D; dark gray bar in the middle), and total benefit (TBEN-D = (EBEN-D) + (SWORS-D; right black bar) for patients with cervical dystonia (CD), other cranial dystonia (CRD), hemifacial spasm (HFS), and migraine (MIG) at the day of recruitment. Standard deviations, indicating the variability of the responses within each disease entity, are presented as error bars.

**Figure 3 jcm-14-00480-f003:**
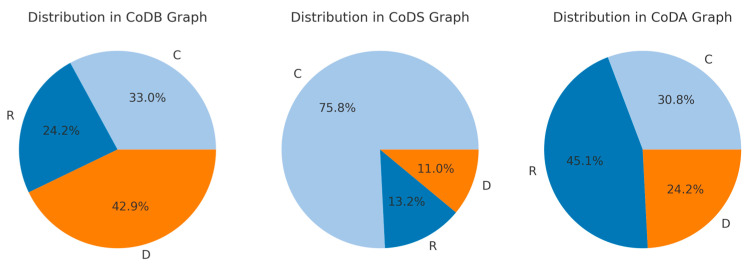
Highly significant (chi^2^-testing: *p* < 0.001) difference in the proportional distributions of three response types (R-, C-, D-type) across the three graph categories (CoDB-, CoDS-, CoDA-graphs). The three pie-charts highlight the relative percentages of R-, C-, and D-response types within the overall CoDB, CoDS, and CoDA graph categories.

**Figure 4 jcm-14-00480-f004:**
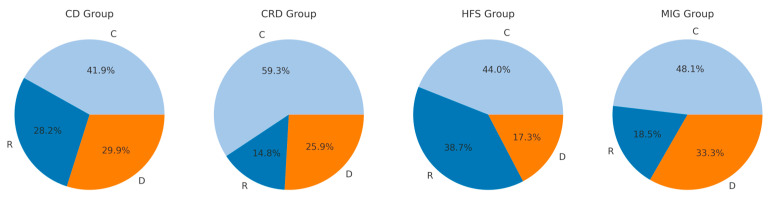
Highly significant (chi^2^-testing: *p* < 0.001) difference in the proportional distributions of the three response types (R-, C-, D-type) across the four disease entities. These four pie-charts illustrate the proportional distribution of R-,C-, and D-types among the four patient groups (cervical dystonia (CD), other cranial dystonia (CRD), hemifacial spasm (HFS), and migraine (MIG).

**Figure 5 jcm-14-00480-f005:**
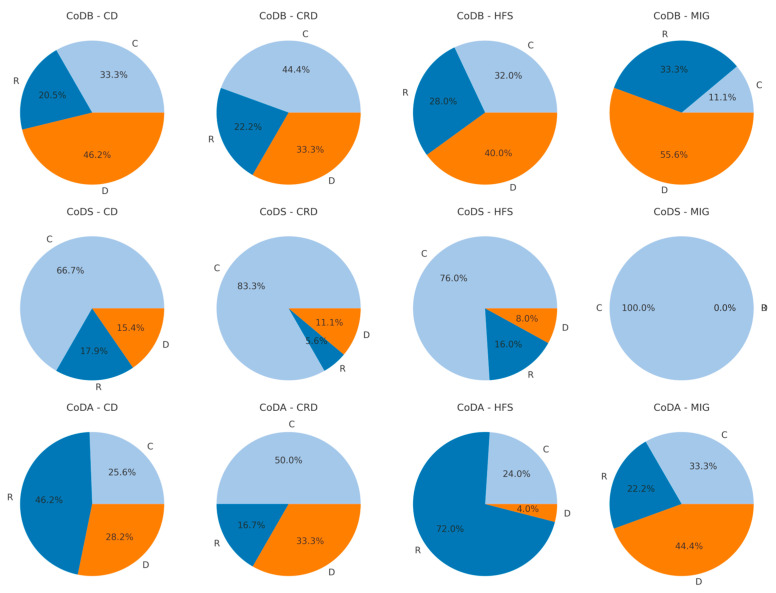
Proportional distributions of the three R-, C-, D-response types across the four disease entities (CD = column 1; CRD = column 2; HFS = column 3; MIG = column 4) for all CoDB-graphs (row 1), all CoDS-graphs (row 2) and all CoDA-graphs (row 3).

**Table 1 jcm-14-00480-t001:** Demographic, treatment related and long-term outcome data of four disease entities.

Parameter	CD	CRD	HFS	MIG
N=	39	19	26	9
f/m	15/24	13/6	12/14	9/0
AGE (yrs)	67.5/12.3	71.4/9.7	73.7/11.7	53.0/8.2
AOS (yrs)	47.8/14.0	58.8/10.2	57.6/11.4	23.4/12.1
AOT (yrs)	52.8/12.4	62.0/9.3	60.5/11.6	47.7/8.5
DURS (yrs)	5.1/5.5	3.1/4.2	2.9/1.8	24.3/14.6
DURT (yrs)	14.7/7.7	9.5/5.4	13.3/7.6	5.3/1.7
RS-%	50.4/39.4	45.3/32.8	40.3/36.8	32.8/15.8
RS-D	53.0/43.8	48.1/34.3	41.2/36.4	31.2/16.9
SS-%	150.8/26.2	164.7/34.2	136.7/30.5	108.9/17.3
SS-D	162.4/35.2	170.0/34.2	143.1/35.4	111.2/22.5
ADOSE (uDU)	337/107	106/69.7	46.6/29.5	160/48.5
abo/ona/inco	10/2/27	8/5/6	17/2/7	0/9/0
STF/P-STF/PTF	8/2/2	0/2/0	0/0/0	0/0/0
ATSUI	3.79/2.27	-	-	-

CD = cervical dystonia; CRD = other cranial dystonia; HFS = hemifacial spasm; MIG = migraine; f = female; m = male; AGE = age at recruitment; AOS = age at onset of symptoms; AOT = age at onset of therapy; DURS = time between onset of symptoms and onset of therapy; DURT = duration of treatment; RS-% = remaining severity (according to questionnaire); RS-D = remaining severity (according to drawing); SS-% = suspected severity (according to questionnaire); SS-D = suspected severity (according to drawing); ADOSE = dose at recruitment; abo/ona/inco = number of patients treated with abo- or ona- or incoBoNT/A; STF/P-STF/PTF = number of patients with secondary treatment failure (STF) or pseudo-STF (P-STF) or primary treatment failure (PTF); ATSUI = TSUI-score [22] at recruitment in patients with CD scores the severity of CD.

**Table 2 jcm-14-00480-t002:** Distributions of the 3 graph-types (R-, C-, D-type) in four patient groups.

Type/Disease	CD (*n* = 39)	CRD (*n* = 18)	HFS (*n* = 25)	MIG (*n* = 9)	ALL (Graphs)
CoDB-R	8	4	7	3	22
CoDB-C	13	8	8	1	30
CoDB-D	18	6	10	5	39
CoDS-R	7	1	4	0	12
CoDS-C	26	15	19	9	69
CoDS-D	6	2	2	0	10
CoDA-R	18	3	18	2	41
CoDA-C	10	9	6	3	28
CoDA-D	11	6	1	4	22
ALL (graphs)	CD (*n* = 117)	CRD (*n* = 54)	HFS (*n* = 75)	MIG (*n* = 27)	ALL (*n* = 273)
Type R	33	8	29	5	75
Type C	49	32	33	13	127
Type D	35	14	13	9	71

R = rapid; C = continuous; D = delayed; CD = cervical dystonia; CRD = (other) cranial dystonia; HFS = hemifacial spasm; MIG = migraine; CoDB = course of disease before BoNT; CoDS = suspected course of disease without BoNT; CoDA = course of disease after BoNT.

## Data Availability

Data are available upon request related to the restrictions of privacy or ethics. The data presented in this study are available upon request from the corresponding author.

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
