# Peer review of "Preventive Effects of Botulinum Neurotoxin Long-Term Therapy: Comparison of the ‘Experienced’ Benefits and ‘Suspected’ Worsening Across Disease Entities"

_jcm, 2025, doi:10.3390/jcm14020480_

Round 1
Reviewer 1 Report
Comments and Suggestions for Authors
My comments and suggestions for authors are:
1. The transition to the study’s specific goals feels abrupt. It would strengthen the introduction if you could clearly articulate the research gap—specifically, why examining "suspected worsening" alongside "experienced benefits" is novel and valuable. This would provide a smoother flow and better context for your study.
2. Consider including some recent meta-analyses or clinical reviews that focus on comparing BoNT’s efficacy across the different disease groups you analyzed. This would help situate your findings within the broader body of research.
3. The methods section could use more detail on how potential biases in patient self-assessments were managed. Additionally, the graph classification criteria (R-, C-, D-types) are subjective and don’t appear to have external validation. Adding information about how recall bias was mitigated and standardizing graph-drawing instructions would be beneficial. Including inter-rater reliability statistics for graph classification would also enhance the robustness of this section.
4. The sections on "experienced improvement" and "suspected worsening" overlap in places, leading to some repetition. Reorganizing these results to avoid redundancy and streamlining the discussion of key findings would improve clarity. Providing concise summaries for each disease group could help readers quickly grasp the main trends.
5. The conclusions are well-supported by the data, particularly the point about the preventive effects of BoNT therapy. However, the statement that patients "assessed their condition realistically" would benefit from additional evidence or clarification. As it stands, this assertion feels somewhat unsubstantiated.
Author Response
|
My comments and suggestions for authors are: 1. The transition to the study’s specific goals feels abrupt. It would strengthen the introduction if you could clearly articulate the research gap—specifically, why examining "suspected worsening" alongside "experienced benefits" is novel and valuable. This would provide a smoother flow and better context for your study. 2. Consider including some recent meta-analyses or clinical reviews that focus on comparing BoNT’s efficacy across the different disease groups you analyzed. This would help situate your findings within the broader body of research.
|
The introduction is revised, and a smoother flow of thoughts is provided.
BoNT´s efficacy across the different disease entities is more emphasized in the revised manuscript. Especially the study by Lee et al. is more frequently be mentioned.
The classification criteria for graph types are based on a mathematical definition and therefore are not subjective. Therefore, no interrater reliability has to be controlled. However, it has to be discussed whether our criteria were reasonable or not. This is now indicated. Indeed, the usefulness of our criteria has to be demonstrated in further studies following the present pilot study.
The discussion has been revised to avoid overlaps between different sections.
In the result section we have already presented the long-term outcome data for each disease entity separately as well as in the discussion.
We have discussed this crucial point of “realistic assessment” for CD-patients for which we have more detailed information. In the revised manuscript we have omitted this problematic closing sentence.
The authors are thankful to reviewer 1 for careful reading and helpful comments.
|
Reviewer 2 Report
Comments and Suggestions for Authors
Dear Authors,
the study assessed the benefit of BoNT therapy in cervical dystonia, hemifacial spasm and the migraine patients. Here my comments:
Here’s the corrected and refined version of your comments:
KEYWORDS: Please consider using MeSH keywords to enhance the discoverability of the article.
ABSTRACT: In the results section, quantify the "benefit" using a significant or non-significant outcome (p-value) if possible. If this is not feasible due to the absence of a scale or scoring system, clarify this limitation. Also, note the repetition of the word "experienced"—please revise for clarity and readability (revised all of the main text).
INTRODUCTION: The reference to Simpson (6) highlights the use of BoNT in treating blepharospasm, cervical dystonia, adult spasticity, and headache. Consider also discussing the outcomes (not always positive) of BoNT in focal hand dystonia (DOI: 10.33588/rn.7208.2020421).
METHODS: Please specify the scales used for assessment, such as Tardieu, Ashworth, or any quality-of-life measurements. Clear identification of the tools used will strengthen the methodology.
TABLES: The legend for ATSUI=STUI-score is unclear. Provide a complete explanation of what this score represents in the text. Additionally, try to minimize the use of abbreviations, even if they are explained in the legend, as it can make the data less accessible to readers unfamiliar with these terms.
Author Response
|
Dear Authors, Here’s the corrected and refined version of your comments: KEYWORDS: Please consider using MeSH keywords to enhance the discoverability of the article. ABSTRACT: In the results section, quantify the "benefit" using a significant or non-significant outcome (p-value) if possible. If this is not feasible due to the absence of a scale or scoring system, clarify this limitation. Also, note the repetition of the word "experienced"—please revise for clarity and readability (revised all of the main text). INTRODUCTION: The reference to Simpson (6) highlights the use of BoNT in treating blepharospasm, cervical dystonia, adult spasticity, and headache. Consider also discussing the outcomes (not always positive) of BoNT in focal hand dystonia (DOI: 10.33588/rn.7208.2020421). METHODS: Please specify the scales used for assessment, such as Tardieu, Ashworth, or any quality-of-life measurements. Clear identification of the tools used will strengthen the methodology. TABLES: The legend for ATSUI=STUI-score is unclear. Provide a complete explanation of what this score represents in the text. Additionally, try to minimize the use of abbreviations, even if they are explained in the legend, as it can make the data less accessible to readers unfamiliar with these terms. |
We also included the disease entity of cranial dystonia. These patients had the highest values for suspected worsening.
We have modified the list of key words.
In the Abstract as well as in the Methods as well as in the results we have now added p-values on the benefit.
We have tried to reduce the inflation of “experienced”.
We have added this paper on hand dystonia.
Because of the special complexity we have not included patients with spasticity. Instead, we now emphasize the use of subjective assessment and CoD-graph drawing in contrast to standard scales.
The TSUI-score is an easy to perform standard scale helping to quantify long-term outcome in cervical dystonia (CD). In our institution the TSUI-score is determined at each treatment session for each CD-patient. This is now explained in detail (we apologize for that).
Authors are also thankful to reviewer 2 for careful reading and helpful comments.
|
Round 2
Reviewer 1 Report
Comments and Suggestions for Authors
I believe the authors have adequately addressed the comments raised during the initial review. I find the manuscript sufficiently improved and suitable for publication in this form.
Reviewer 2 Report
Comments and Suggestions for Authors
Dear Authors,
I appreciated your efforts in improving the manuscipt. It results interesting and clearer.